# Centrifugation-Assisted Solid-Phase Extraction Coupled with UPLC-MS/MS for the Determination of Mycotoxins in ARECAE Semen and Its Processed Products

**DOI:** 10.3390/toxins14110742

**Published:** 2022-10-29

**Authors:** Huanyan Liang, Qianyu Hou, Yakui Zhou, Lei Zhang, Meihua Yang, Xiangsheng Zhao

**Affiliations:** 1Hainan Provincial Key Laboratory of Resources Conservation and Development of Southern Medicine & Hainan Branch of the Institute of Medicinal Plant Development, Chinese Academy of Medical Sciences & Peking Union Medical College, Haikou 570311, China; 2Institute of Medicinal Plant Development, Chinese Academy of Medical Sciences & Peking Union Medical College, Beijing 100193, China; 3School of Chinese Materia Medica, Guangdong Pharmaceutical University, Guangzhou 510006, China

**Keywords:** multi-mycotoxins, SPE, *Areca catechu*, UPLC-MS/MS

## Abstract

Mycotoxins can occur naturally in a variety of agriculture products, including cereals, feeds, and Chinese herbal medicines (TCMs), via pre- and post-harvest contamination and are regulated worldwide. However, risk mitigation by monitoring for multiple mycotoxins remains a challenge using existing methods due to their complex matrices. A multi-toxin method for 22 mycotoxins (aflatoxin B_1_, B_2_, G_1_, G_2_, M_1_, M_2_; ochratoxin A, B, C; Fumonisin B_1_, B_2_, B_3_; 15-acetyldeoxynivalenol, 3-acetyldeoxynivalenol, diace-toxyscirpenol, HT-2, T-2, deepoxy-deoxynivalenol, deoxynivalenol, neosolaniol, zearalenone, and sterigmatocystin) using centrifugation-assisted solid-phase extraction (SPE) clean-up prior to ultra-performance liquid chromatography-tandem mass spectrometry (UHPLC-MS/MS) analysis for Arecae Semen and its processed products was developed and validated. Several experimental parameters affecting the extraction and clean-up efficiency were systematically optimized. The results indicated good linearity in the range of 0.1–1000 μg/kg (*r*^2^ > 0.99), low limits of detection (ranging from 0.04 μg/kg to 1.5 μg/kg), acceptable precisions, and satisfactory recoveries for the selected mycotoxins. The validated method was then applied to investigate mycotoxin contamination levels in *Areca catechu* and its processed products. The mycotoxins frequently contaminating *Areca catechu* were aflatoxins (AFs), and the average contamination level and number of co-occurring mycotoxins in the Arecae Semen slices (Binlangpian) were higher than those in commercially whole Arecae Semen and Arecae Semen Tostum (Jiaobinlang). Sterigmatocystin was detected in 5 out of 30 Arecae Semen slices. None of the investigated mycotoxins were detected in Arecae pericarpium (Dafupi). The results demonstrated that centrifugation-assisted SPE coupled with UHPLC-MS/MS can be a useful tool for the analysis of multiple mycotoxins in *Areca catechu* and its processed products.

## 1. Introduction

Mycotoxins are the secondary metabolites of fungal origin produced by various genera (*Aspergillus*, *Fusarium*, *Penicillium*, etc.) and have been a global concern for their ability to induce acute to chronic toxicity in humans and animals due to their teratogenic, mutagenic, carcinogenic, immunosuppressive, and endocrine-disrupting effects [1]. Up to now, more than 400 mycotoxins have been identified with various chemical structures. Mycotoxin contamination can occur in different matrices, such as foods, cereals, feeds, and traditional Chinese medicines (TCMs), as well as related products [2,3,4]. In these matrices, several mycotoxin groups are frequently found, including aflatoxins, ochratoxin, fumonisins, zearralenone, T-2/HT-2, and deoxynivalenol. Despite efforts to control fungal infection, extensive mycotoxin contamination has been reported in foods and TCMs.

*Areca catechu* Linn, an evergreen areca tree in the palm family, is widely distributed and cultivated in several tropical southern and southeastern Asian countries such as China, India, Bangladesh, Indonesia, Myanmar, Thailand, Malaysia, Vietnam, and the Philippines [5]. The main producing areas in China are Hainan, Guangdong, Guangxi, Yunnan, and Taiwan provinces. Arecanut palm is a popular crop in the south and southeast of Asia for both social and economic reasons. Arecae Semen is an addictive substance that is widely used as a mouth freshener and to assist digestion. As a traditional herb medicine, Arecae Semen has been used to kill parasites and promote digestion, with over 100 prescriptions containing Arecae semen Semen and its processed products [6]. The most common clinically used forms of *Areca catechu* are Arecae Semen, Arecae Semen Tostum, and Arecae pericarpium. Arecae Semen (Binglang in Chinese, BL) is the dried ripe seed of *Areca catechu*. Slices of Arecae Semen are stir-baked until a charred yellow color appears to obtain Arecae Semen Tostum (Jiaobinglang in Chinese, JBL). Arecae pericarpium (Dafupi in Chinese, DFP) is the dried pericarp of *Areca catechu* (Figure 1). Arecae Semen is susceptible to contamination by pathogenic fungi during various stages of harvesting, storage, and transportation. Furthermore, Arecanut palm is frequently cultivated in countries with high temperature and humidity climates [7]. Because of these conditions, as well as post-harvest environmental factors, fungi are capable of producing mycotoxins in areca products and medicines. The safety of the areca nut has gradually captured attention due to its extensive medicinal and edible value.

Several LC-MS/MS (or fluorescence detection, FLD) methods have been reported for the analysis of mycotoxins in *Areca catechu*. However, none so far have focused on the occurrence and detection of mycotoxin residues in *Areca catechu*-processed products [7,8,9]. Hongmei Liu et al. reported the development of a UFLC-MS/MS based on a one-step extraction without any further clean-up for the quantitative analysis of 11 mycotoxins in *Areca catechu* [8], whereas Muhammad Asif Asghar et al. described an HPLC-FLD method for the simultaneous analysis of aflatoxins (AFs, AFB_1_, AFB_2_, AFG_1_, AFG_2_) in betel nuts, and compared the level of contamination amongst Asian countries. Samples were extracted with a mixture of ACN/H_2_O, followed by immunoaffinity column clean-up [7]. Hung-Yu Lin et al. developed an HPLC-MS/MS to determine nine mycotoxins in the betel nut [9]. Despite satisfactory recovery and repeatability, these methods included only a limited number of mycotoxins, involved significant matrix effects, or required complex pretreatments that are time-consuming and can result in inaccurate results. Low levels of mycotoxins and complex matrices of Arecae Semen and its processed products require an efficient clean-up method to improve the sensitivity and selectivity of the analytical method. Solid-phase extraction is one of the most commonly used methods to remove matrix interferences. Traditionally, during the wash, eluent solvent passes through the SPE column by gravity or negative pressure. However, this tedious procedure increases the analysis time or requires vacuum pump equipment, leading to a reduction in the number of samples that can be processed at one time. Hence, it is necessary to develop a rapid and sensitive approach to detect the presence of multi-mycotoxins in Arecae Semen and its processed products.

Hence, the present study describes a simple and rapid quantification method that was developed based on centrifugation-assisted SPE purification coupled with UPLC-MS/MS to accurately detect and quantify 22 mycotoxins (aflatoxin B_1_ (AFB_1_), aflatoxin B_2_ (AFB_2_), aflatoxin M_1_ (AFM_1_), aflatoxin M_2_ (AFM_2_), aflatoxin G_1_ (AFG_1_), aflatoxin G_2_ (AFG_2_), 15-acetyldeoxynivalenol (15-Ac DON), 3-acetyldeoxynivalenol (3-Ac DON), diacetoxyscirpenol (DAS), HT-2, T-2, deepoxy-deoxynivalenol (DOM-1), deoxynivalenol (DON), neosolaniol (NEO), zearalenone (ZEN), sterigmatocystin (ST), ochratoxin A (OTA), ochratoxin B (OTB), ochratoxin C (OTC), fumonisin B_1_ (FB_1_), fumonisin B_2_ (FB_2_), and fumonisin B_3_ (FB_3_)) in Arecae Semen and its processed products. To investigate the contamination levels of target mycotoxins, the proposed method was successfully applied to 75 batches of Arecae Semen and its processed products, which were obtained from different markets and stores in China.

## 2. Results and Discussion

### 2.1. Optimization of the UPLC-MS/MS System

Careful optimization of the UPLC-MS/MS parameters was achieved in order to obtain suitable separation and better sensitivity for target mycotoxins. An initial investigation was carried out to optimize the MS parameters for mycotoxin detection. A standard solution of 1.0 μg/mL of each mycotoxin was injected directly into the MS using flow injection, and a full scan was executed in the first quadrupole in electrospray ionization (ESI) positive and negative modes. The results showed that 22 target mycotoxins had better responses in the positive ion mode. For most mycotoxins (AFs, OTS, FBs, etc.), there is a high abundance of the [M + H]^+^ peak in positive ESI mode since they contain methyl or carbonyl groups. The [M + NH_4_]^+^ ions among the precursor of T-2 and HT-2 showed the highest ion abundance. The high DAS ion abundance was in the [M + Na]^+^ ion form. The signal acquisition information of each mycotoxin in the optimized multiple reaction monitoring (MRM) mode, including characteristic ion pairs, fragmentor voltage, and collision energies, are listed in Table 1. For each mycotoxin, two mass transitions with the highest abundances were selected. One quantifier ion and one qualifier transition were monitored for each mycotoxin, whereas only one transition was chosen for the internal standard (IS). Finally, the MRM experiments were segmented according to the separation time of target mycotoxins with the goal of increasing the sensitivity of the analysis since the MS scans only these ions during that specific time interval. The higher specificity and sensitivity obtained when employing MRM segmentation, and multiple transitions are expected to result in better quantification [10].

Because of the diverse structure and physio-chemical properties of mycotoxins, it is important to optimize the chromatography column and the composition of the mobile phase. Based on previous experience, a Waters Acquity BEH C_18_ column (100 mm × 2.1 mm, 1.7 μm, Waters) was selected as the separation column for the current application. The mobile phase was further evaluated, including both organic (MeOH and ACN) and aqueous phases with different buffer solutions (acetic acid (AA), formic acid (FA), ammonium formate, and ammonium acetate). For all mycotoxins, a better response was observed when using the MeOH/H_2_O_2_ as a mobile phase compared to ACN/H_2_O_2_. A similar report pointed out that this trend might be attributed to the protic nature of MeOH, which enhances the response of [M + H]^+^ in ESI^+^ mode [11]. An acidic mobile phase could improve the signal intensities for some mycotoxins (FBs, OTA) due to the presence of carboxylic groups in the molecular structure [12]. Moreover, the acidification of the mobile phase with ammonium acetate could promote T-2 and HT-2 to form ammonium adducts [M + NH_4_]^+^ as precursor ions. Additionally, it could avoid the formation of unwanted adducts (e.g., [M + Na]^+^) [13]. When the mobile phase was fixed as 2 mM ammonium acetate and MeOH (containing 0.1% FA), all mycotoxins were separated with resolved symmetrical peaks and at high abundances. Other parameters, including column temperature, flow rate, and injection volume, were also optimized. The MRM chromatograms of the target mycotoxins under the optimized UPLC-MS/MS conditions are shown in Figure 2.

### 2.2. Optimization of Sample Preparation

For the detection of multiple mycotoxins, the method of extraction and clean-up is a crucial step prior to UPLC-MS/MS analysis. An excellent extraction capability could be considered the most important factor in the selection of the extraction solvent. MeOH and acetone were not considered due to the insolubility of the mycotoxins in MeOH and the unselective solvent power of acetone [14]. ACN was chosen because it has been shown to extract the broadest range of organic compounds without the co-extraction of large amounts of lipophilic material [15]. However, when ACN was used as the extraction solvent, the extraction effect of mycotoxins sensitive to the polarity range was not ideal. Adding acid to the extraction solution can improve the extraction rate of some acid toxins (such as FBs, OTA, and T-2) and AFs [16]. Therefore, the extraction recoveries of mycotoxins by ACN (80% and 84% aqueous ACN) with different concentrations of FA were evaluated by spiking blank samples with target mycotoxins. Figure 3 shows that the recovery of most of the mycotoxins extracted with 84% aqueous ACN was better than using 80% aqueous ACN. In addition, when FA was added to the extraction solvent, it resulted in increased recovery of some target mycotoxins (OTA, OTB, OTC, DON, DAS, HT-2, T-2, and NEO). It is important to note that when the concentration of FA in the extraction solvent is above 0.3%, there will be reduced recovery of some toxins, such as AFs, OTA, OTB, OTC, DOM-1, DON, 15-AcDON, 3-AcDON, etc. Thus, ACN with 0.2% FA was selected for further experiments. Next, the effects of the volume of extraction agent (10, 15, and 20 mL) and extraction time (5, 10, and 15 min) were evaluated. When the volume of the extraction solvent was above 15 mL, there was no significant increase in the recovery of the mycotoxins. As for the extraction time, it can be highlighted that the recovery of the mycotoxins increased from 5 to 10 min and then decreased or kept constant. Using the proposed extraction method with these conditions achieved satisfactory extraction efficiency with recoveries ranging from 67.1 to 111.2%.

Matrix effects caused by the co-extraction of compounds from the matrix are a known limitation of LC-MS/MS analysis. An efficient clean-up procedure following extraction can serve to decrease matrix effects. In recent years, multi-functional purification columns have been popular in LC-MS/MS preprocessing for the simple and fast removal of impurities [17]. In our preliminary research, SPE (MycoSpin^TM^ 400) was more effective and convenient compared with the other clean-up pretreatments. Thus, the centrifugation-assisted SPE method was selected as the clean-up method in this study. In what follows, two parameters that affected the extraction recoveries, pH and load volume of sample extraction solvent, were carefully optimized. BLS was used as a blank matrix, and the test sample was prepared by spiking in a standard mixture (5.0 μg/kg for AFB_1_, AFB_2_, AFG_1_, AFG_2_, AFM_1_, and AFM_2_; 25.0 μg/kg for OTA, OTB, and OTC; 50.0 μg/kg for DON, DOM-1, HT-2, T-2, ST, ZEN, 15-Ac DON, 3-Ac DON NEO, and DAS; and 125.0 μg/kg for FB_1_, FB_2_, and FB_3_) into the blank. The purification effect was investigated using the recovery rate as the index. An evaluation of the pH adjustment with AA, prior to clean-up, was performed. In order to be convenient for sample testing, the concentration of AA (%, *v*/*v*) was used for optimization instead of the pH value. The recoveries of FB_1_, FB_2_, and FB_3_ were 0 when using the sample extraction solvent (pH = 4.66) without adding AA. Alongside with FBs, the recoveries of OTA, OTB, AFM_2_, AFG_2_, etc., were below 40%. Figure 4 shows the effect of AA concentration on the recovery of mycotoxins after clean-up. It can be observed that the recoveries of FBs, OTA OTB, OTC, AFM_2_, and AFG_2_ increased significantly when the concentration of AA ranged from 0.5% (pH = 4.6) to 15% (pH = 3.15). In particular, increasing the AA concentration gradually increased the recoveries of NEO, 15-AcDON, 3-AcDON, DAS, T-2, and ST, approaching 100%, although a concentration of AA above 15% (20%, pH = 2.8; 25%, pH = 2.48) did not yield a significant increase in the recoveries of most mycotoxins. However, there was a significant decrease in the recoveries of DON, DOM-1, 15-AcDON, 3-AcDON, DAS, ZEN, and ST. Additionally, the influence of AA concentration on the purification efficiency was investigated by comparing the color of the purified extract. Increasing the concentration of AA caused the color of the purified extract to become dark (Appendix A), which indicated that the concentration of acid affected the binding ability to adsorb and the co-extractive components in the matrix. It was therefore decided to maintain the concentration of AA at 15% in order to obtain maximum recovery. Hence, the pH value of the loading solution was a key factor affecting the accuracy of the SPE method. For WBL, JBL, and DFP, the trend for color change in the purified solution with different acid concentrations was similar to that of BLS. In addition, after adding different concentrations of AA, the pH values of the loading solvents from different samples (WBL, JBL, and DFP) were not significantly different from that of BLS (within ±0.25). Under the optimized conditions, the recoveries of most selected mycotoxins in WBL, JBL, and DFP ranged from 70% to 120% (Appendix A), which indicated that the developed clean-up method is suitable for these herbs.

As we know, high concentrations of acid in the mobile phase and sample extraction solvent would inhibit MS response for all analytes. To investigate the effectiveness of AA on the target mycotoxins in the present study, a mycotoxin-free BLS sample was used. The sample preparation method was the same as detailed above. After SPE purification, one sample was evaporated to dryness under a gentle nitrogen steam and reconstituted with an ACN/H_2_O_2_/FA (84/15.8/0.2, *v*/*v*/*v*) solution (Sa), and the other sample was not evaporated with nitrogen (Sb). The MS signal responses of mycotoxins in both solvents were compared using the following equation: IA (%) = (1 − Sa/Sb) × 100. As shown in Figure 5, the addition of AA (15%) had different inhibitory effects on the MS signal response of selected mycotoxins. The inhibition rate was in the range of 16.76% (DAS) ~ 45.19% (ST). Therefore, after the sample was purified by SPE, the acid was removed with nitrogen from the eluent.

Next, different load volumes (300, 450, 600, 750, and 900 μL) of extract were evaluated to estimate the capacity of saturation of the solid phase. The recovery rates shown in Appendix A indicated that acceptable recovers were obtained with a load volume of 750 μL. Therefore, this load volume was chosen as the optimum volume. Finally, the rotation speed and time of centrifugation were evaluated and shown to have no significant effect on the recovery of the target mycotoxins.

### 2.3. Matrix Effect

In mycotoxin analysis, the matrix effect (ME) is often observed when ESI is used as the ionization technique in LC-MS/MS. The ion intensity of target analytes is suppressed or enhanced by co-extractive compounds coming from the matrix [18,19]. ME affects the reproducibility and accuracy of the proposed LC-MS/MS method, which is considered one of the major drawbacks. In our team’s previous research, we observed that BLS has a significant ME on some mycotoxins [8]. To evaluate the possible occurrence of ME, the MS/MS peak area of the mycotoxins spiked in the extraction solvent was compared to those spiked in mycotoxin-free sample extraction solvent after SPE clean-up at the same concentration level. In general, ME is considered to be tolerable with a value in the range between 80~120% [20]. Values outside of this range are considered to have strong signal suppression or enhanced effect. As shown in Table 2, the ME of WBL was acceptable, with values for most toxins in the range of 80~120%, except for DON (78.30%), NEO (128.25%), 15-AcDON (78.98%), AFG_2_ (77.91%), AFG_1_ (71.68%), DAS (124.47%), and FB_1_ (109.10%). Although the ME values of these seven toxins were outside the acceptable range, they were very close to the critical value. Similar ME values were found in BLP and JBL. However, for DFP, six mycotoxins (DOM-1, AFG_2_, AFG_1_, FB_1_, FB_2_, and FB_2_) showed a strong suppressive effect in the range of 56.50~79.97%. Apparently, ME still remained high for a small number of toxins (especially in DFP), although SPE clean-up had been applied to remove co-extracted components. The use of stable isotopically labeled internal standards is one of the commonly employed strategies to compensate for the bias caused by ME. Therefore, the quantification of multi-mycotoxins in Arecae Semen and its processed products was carried out using the internal standard curve method, even if strong ME was only noticed in DFP.

### 2.4. Method Validation

Due to its selectivity, this analytical method was chosen to evaluate obvious sources of co-extractive compounds at definitive retention times in blank samples. The retention times of the target mycotoxins in the chromatograms of the blank samples showed high levels of repeatability without interfering peaks, indicating that this approach has high selectivity. The comprehensive characteristic performance parameters are listed in Table 3. Linear calibration curves were obtained over the range of 0.1–1000 μg/kg for the target mycotoxins with linear regression coefficients (*r*^2^) exceeding 0.99. The limits of detection (LOD) and limits of quantification (LOQ) values were in the range of 0.04–1.5 μg/kg and 0.1–5.0 μg/kg, respectively. The precisions of intra- and inter-day were lower than 8%. The average recoveries of mycotoxins at three different concentration levels ranged from 70% to 120%, with the exception of DON and DOM-1, and the relative standard deviation (RSD) values were mostly <15.0% (Table 4). These validation data indicated that the developed method was appropriate for the simultaneous determination of 22 mycotoxins in BL with satisfactory performance according to the European Commission regulations [21].

### 2.5. Application

The proposed SPE-UPLC-ESI-MS/MS method was applied to determine the target mycotoxins in BL and its processed products purchased from local markets. Each sample was prepared in triplicate for extraction, clean-up, and LC-MS/MS analysis under the optimized conditions. The results of the mycotoxin contamination in the WBL, BLS, and JBL samples are presented in Table 5. The MRM chromatogram of the detected mycotoxins in the different positive samples is shown in Appendix A. None of the DFP collected was contaminated with the selected mycotoxins. AFs were detected in the WBL, BLS, and JBL samples. This result was similar to that found in a previous study [8], in which large amounts of AFs were detected in BLS with an incidence of 16.7%. In addition, ST was detected in 5 BLP samples.

The analyzed results showed a low rate (13.3%, 2 out of 15 samples) of mycotoxin detected in WBL. As shown in Table 5, three AF mycotoxins (AFB_1_, AFB_2_, and AFM_2_) were detected in the two positive WBL samples (WBL-4 and WBL-11). The concentrations of AFB_1_, AFB_2_, and AFM_2_ were 5.43 μg/kg, 2.78 μg/kg, and Tr (<LOQ), respectively, in the WBL-4 sample. It was observed that the level of AFB_1_ in WBL-4 exceeded the MRLs (5.0 μg/kg) established for AFB_1_ in Chinese Pharmacopoeia (Ch. P) [22]. For another positive sample (WBL-11), the concentration of AFs was lower than the value of LOQ. In the BLS samples, there was a higher frequency of occurrence for AFs (43.3%) as well as a higher level of contamination when compared with that of WBL. The level of AFs in the positive samples ranged from Tr to 7.55 μg/kg. The highest level of AFB_1_ was determined to be 7.55 μg/kg. Furthermore, ST at a concentration range of Tr to 2.17 μg/kg was found in 16.7% (5 out of 30) of the samples. In general, the amount of AFB_1_ and total AFs in the BL-2 sample exceeded the MRLs regulated by ChP. In the 15 JBL samples, five AFs mycotoxins were detected in six samples (40%), with concentrations ranging from Tr to 4.02 μg/kg.

ST is a polytetide mycotoxin produced by some fungal species, including Aspergillus flavus, A. Parasiticus, A. Versicolos, etc. As a precursor of AFB_1_, ST has a similar molecular structure to AFB_1_ and shares its biosynthetic pathway [23]. ST is considered to be a potential carcinogen, mutagen, and teratogen and has been classified as a group 2B chemical [24]. However, there is limited information available in the literature on the presence of ST in herbs and their related products [25]. In this study, the co-occurrence of ST-AFB_1_ was observed in five samples with low contamination levels of ST (≤2.17 μg/kg). This may be related to the contaminated fungal species. In aflatoxigenic species, in which ST is converted into AFB_1_, the accumulation of ST is rarely seen. Although, some species (A. versicolor, A. nidulans, etc.) are unable to convert ST into AFB_1_, probably due to the lack of genes encoding the specific methyltransferase required for this conversion [26]. For this reason, substrates colonized by these fungi can contain high amounts of ST. Therefore, it can be speculated that the species contaminated with areca nut may be Aspergillus flavus or A. parasiticus. In these fungi, ST does not accumulate in large amounts since it is consumed to produce AFB_1_.

A previous report found that the fungal counts of A. flavus in WBL, BLS, and JBL were significantly different (BLS > JBL > WBL) [27]. This information indicated that these results are closely related to the consequence of the present study. Additionally, the components and processing method may affect fungal growth. The areca nut is rich in polysaccharides, fatty acids, and lipids, which may be beneficial for fungal growth and the production of toxins. Alternatively, DFP contains a large amount of fiber, and triterpenes in the DFP possess significant antifungal activity [28]. Furthermore, the processing itself may destroy aflatoxins or may cause chemical changes that render boiled and baked nuts less favorable substrates for fungal growth. Consequently, to minimize the risk of mycotoxin contamination in *Areca catechu* and its processed products, attention could be paid to specific factors, such as suitable culture management, appropriate drying, processing procedures, and storage conditions.

## 3. Conclusions

In this study, a centrifugation-assisted SPE coupled with UHPLC-MS/MS was developed and validated for the fast and sensitive detection of multi-mycotoxins in *Areca catechu* and its processed products. This method could be applied for the qualitative and quantitative analysis of the 22 selected mycotoxins in herbs, providing quality and safety to the herb industry and consumers. However, it should be emphasized that the pH value of the extract prior to SPE purification may be adjusted according to the tested TCMs. Mycotoxin analysis showed that *Areca catechu* is susceptible to AF contamination caused by inadequate harvesting and storage conditions. Therefore, continuous monitoring for multiple mycotoxins by the proposed method is important for confirmation of current contamination and further assurance of the safe consumption of *Areca catechu* and its related products. Considering the toxicity of ST and its involvement in the biosynthesis of AFB_1_, setting a limit for ST in TCMs is essential to ensure their safety.

## 4. Materials and Methods

### 4.1. Chemicals and Reagents

HPLC grade MeOH, ACN, ammonium formate, FA, and AA were obtained from Thermo Fisher Scientific (Waltham, MA, USA). Deionized water was prepared using a Milli-Q system (Millipore Corporation, Bedford, MA, USA). The MycoSpin^TM^ 400 SPE cartridge was purchased from Romer Labs (Tulln, Austria). Certified standard solutions of AFB_1_ (25 μg mL^−1^), AFB_2_ (25 μg mL^−1^), AFM_1_ (10 μg mL^−1^), AFM_2_ (10 μg mL^−1^), AFG_1_ (25 μg mL^−1^), AFG_2_ (25 μg mL^−1^), 15-AcDON (100 μg mL^−1^), 3-AcDON (100 μg mL^−1^), DAS (, 100 μg mL^−1^), HT-2 (100 μg mL^−1^), T-2 (100 μg mL^−1^), DOM-1 (50 μg mL^−1^), DON (50 μg mL^−1^), NEO (100 μg mL^−1^), ZEN (25 μg mL^−1^), ST (50 μg mL^−1^), OTA (10 μg mL^−1^), OTB (10 μg mL^−1^), OTC (10 μg mL^−1^), FB_1_ (50 μg mL^−1^), FB_2_ (FB_2_, 50 μg mL^−1^),FB_3_ (FB_3_, 50 μg mL^−1^), [^13^C_17_]-AFB_1_ (0.5 μg mL^−1^), [^13^C_20_]-OTA (10 μg mL^−1^), and [^13^C_34_]-FB_1_ (25 μg mL^−1^) were acquired from Romer Labs (Butzbach, Germany) with purities above 98%.

### 4.2. UPLC-MS/MS Conditions

A Waters Acquity UPLC-tandem quadrupole (TQD) mass spectrometer (Waters, Milford, MA, USA) with ESI was used for detection. The target mycotoxins were separated on an Acquity UPLC BEH C_18_ (1.7 µm, 2.1 mm × 100 mm) column with the column temperature set at 40 °C. The mobile phase was comprised of methanol containing 0.1% FA as eluent A and 2 mM ammonium formate in water as eluent B. The gradient elution system was as follows: 0 min: 25% A; 1 min: 25% A; 4 min: 45% A; 6 min: 60% A; 14 min: 95% A; 19 min: 95% A; 20 min: 25% A. The flow rate was set at 0.3 mL min^−1^, and the injection volume was 2 μL. The system was re-equilibrated with 25% eluent A for 5 min prior to the next injection. The detection and qualitative analysis were operated in positive electrospray ionization mode with MRM scanning mode. Nitrogen was used as the cone and desolvation gas, and argon was used as the collision gas. The MS conditions were as follows: capillary voltage, 3.5 kV; desolvation temperature, 350 °C; source temperature, 150 °C; desolvation gas, 650 L h^−1^; cone nitrogen gas, 30 L h^−1^. The data were acquired using the software program MassLynx 4.1. The optimal collision energies and cone voltages for each transition of the target mycotoxins are summarized in Table 1. Data acquisition was performed under time-segmented conditions based on the chromatographic separation of the target mycotoxins to maximize detection sensitivity. A segment at 0–6 min was detected for DON, DOM-1, NEO, 3-AcDON, 15-AcDON, and AFM_2_; a segment at 4.5–7.5 min was detected for AFG_2_, AFM_1_, AFG_1_, AFB_2_, AFB_1_, and DAS; and a segment at 7.5–12 min was detected for HT-2, OTB, FB_1_, T-2, OTA, FB_3_, ZEN, ST, FB_2_, and OTC.

### 4.3. Samples

A total of 75 samples of Arecae Semen and its processed products (including 15 WBL, 30 BLS, 15 JBL, and 15 DFP) were analyzed for multi-mycotoxins content. All samples were obtained from pharmacies or markets in the provinces of Hainan, Guangxi, Guangdong, and Yunnan in China P. R. and identified by the authors. Samples were homogenized using a laboratory mill and sieved through a 0.85 mm mesh filter before they were finally stored at −20 °C until the time of the experimental analysis.

### 4.4. Sample Preparation

Homogenized powder from each sample was accurately weighed (2.00 g) and placed in a 50 mL centrifuge tube with 15 mL of ACN/H_2_O_2_/FA (84/15.8/0.2, *v*/*v*/*v*) solution, followed by vortex mixing for 1 min and ultrasonic extraction at room temperature for 10 min. The solution was cooled to room temperature and adjusted to the original weight with the ACN/H_2_O_2_/FA (84/15.8/0.2, *v*/*v*/*v*) solution and then centrifuged at 4000 rpm for 10 min. Subsequently, 150 µL of AA was added to 850 µL of the supernatant and mixed by vortex for 30 s. Then, 750 µL of the mixture was transferred into the MycoSpin^TM^ 400 column, which was capped, vortexed for one minute, and turned upside down before breaking the bottom tip off and placing the column in the centrifuge tube for centrifugation at 2000 rpm for 2 min. Last, 500 µL of the eluting solvent was transferred into 1.5 mL dark vials, and aliquots were evaporated to dryness under a gentle nitrogen steam at 40 °C, reconstituted to 500 µL with the extraction solvent, and filtered through a 0.22 µm PTFE syringe filter for UPLC-MS/MS analysis.

### 4.5. Preparation of Standard Solutions

Working solutions were prepared by diluting the individual stock standard solutions in ACN. The mixed stock solutions of each target mycotoxin were at the following concentrations: 0.25 μg mL^−1^ for AFB_1_, AFB_2_, AFG_1_, AFG_2_, AFM_1_, and AFM_2_; 1.0 μg mL^−1^ for DON, DOM-1, OTA, OTB, OTC, HT-2, T-2, ST, and ZEN; 2.0 μg mL^−1^ for 15-Ac DON, 3-Ac DON NEO, and DAS; and 2.0 μg mL^−1^ for FB_1_, FB_2_, and FB_3_. The standard mixtures were prepared in ACN, stored at −20 °C, and renewed every 2 months.

### 4.6. Method Validation

Method validation was carried out based on the regulation guidelines of the European Commission [21] in terms of the determination of selectively, linearity, LOD, LOQ, precision (intra and inter-day variability), and accuracy. Linearity was evaluated through the coefficient of determination (*r*^2^) of the calibration curves using the internal standard method. The LOD and LOQ were determined as the concentration of the mycotoxin above a signal-to-noise ratio (S/N) of 3 and 10 times, respectively. Intra-day precision was assessed by carrying out six replicates of the working solution in one day and calculating the RSD of the six values. The inter-day precision was determined by measuring the working solutions on three successive days. To evaluate the stability of the analytes in the blank sample extracts, spiked samples were extracted, and mycotoxin concentrations were determined at 0, 2, 4, 8, 12, 16, and 24 h under the optimized UPLC-MS/MS conditions. The results were expressed as RSD for peak areas of the target mycotoxins. Accuracy was determined by spiking a mixed standard solution into the blank sample at three different levels. The spiked samples were extracted and analyzed by UPLC-MS/MS as described above. The recovery was evaluated using the following formula: Recovery (%) = (C_2_/C_1_) × 100, where C_1_ is the fortification concentration of the mycotoxin and C_2_ is the measured concentration of the mycotoxin standard in the blank sample.

### 4.7. Matrix Effect Evaluation

The ME was evaluated by comparing the peak responses of the mycotoxin standards in pure solvent to those in the spiked extract at three levels. The ME was calculated using the following formula: ME (%) = (A_2_/A_1_) × 100, where A_1_ is the average peak area of the mycotoxin standard in the blank solvent (ACN) at a specific concentration, and A_2_ is the average peak area of the mycotoxin standard in the blank extract sample at the same concentration. With an acceptable compliance interval between 80% and 120%, the ME can be ignored. Otherwise, when the values were below 80%, it indicated a substantial ion suppression effect, and when the values were above 120%, it indicated a substantial ion enhancement effect [29].

## Figures and Tables

**Figure 1 toxins-14-00742-f001:**
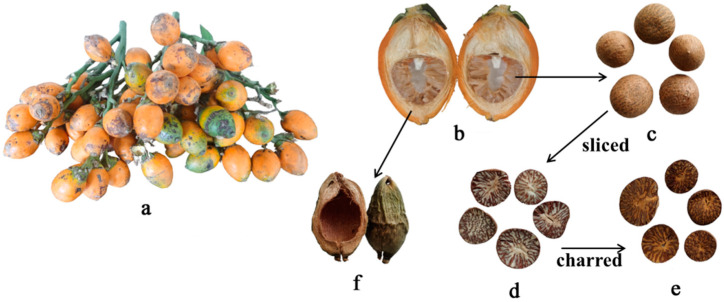
(**a**,**b**) The fruit of *Areca catechu*. (**c**) The whole (WBL) and (**d**) slices (BLS) of Arecae Semen. (**e**) The Arecae Semen Tostum (Jiaobinlang, JBL) and (**f**) Arecae pericarpium (Dafupi, DFP).

**Figure 2 toxins-14-00742-f002:**
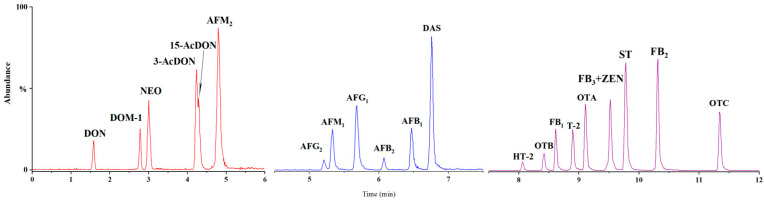
UPLC-MS/MS MRM chromatograms of selected mycotoxins.

**Figure 3 toxins-14-00742-f003:**
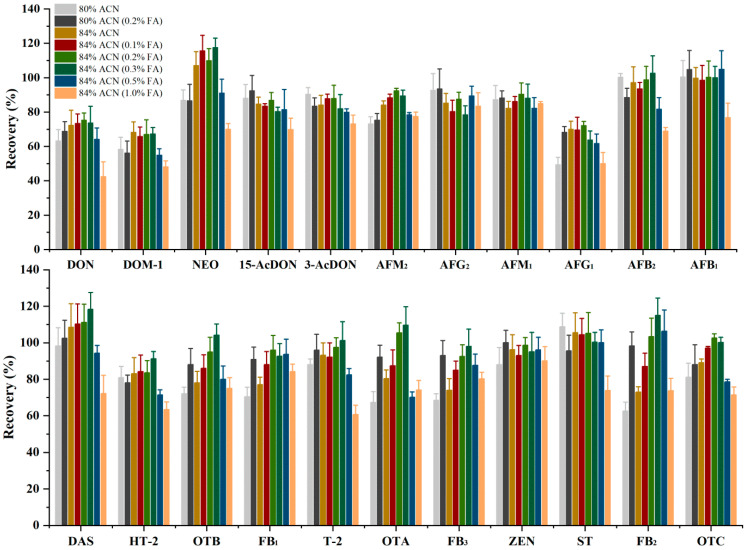
Effects of extraction solvent on the recoveries of 22 mycotoxins.

**Figure 4 toxins-14-00742-f004:**
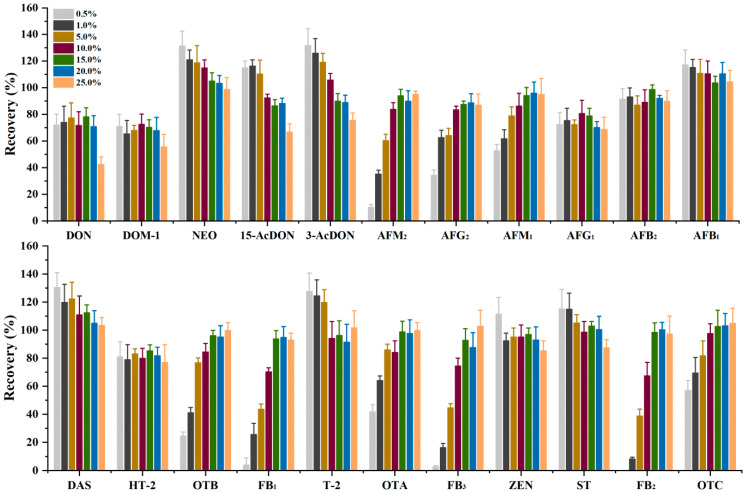
Concentration of AA on the recoveries of selected mycotoxins.

**Figure 5 toxins-14-00742-f005:**
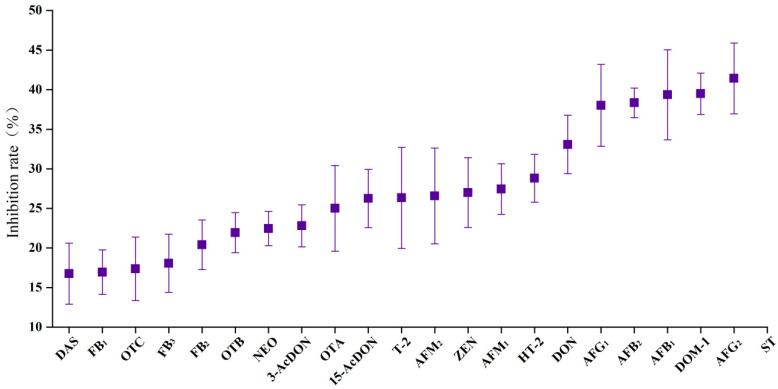
Inhibition rate of AA (15%) on response of target mycotoxins.

**Table 1 toxins-14-00742-t001:** LC-MS/MS (ESI^+^) parameters for 22 mycotoxins.

Mycotoxins	*t_R_* (min)	Parent Ion	Cone Voltage (V)	MRM Transitions			
				Quantification		Identification	
				Product Ion	Collision Energy (eV)	Product Ion	Collision Energy (eV)
DON	1.61	297.1	23	249.1	10	231.1	12
DOM-1	2.77	281.1	22	215.1	13	233.1	13
NEO	3.05	400.37	26	305.14	12	215.1	16
3-AcDON	4.26	339.2	24	231.1	13	203.0	14
15-AcDON	4.34	339.2	24	321.1	10	261.0	12
AFM_2_	4.81	331.1	38	313.1	17	285.1	25
AFG_2_	5.21	331.2	54	245.1	28	189.0	44
AFM_1_	5.33	329.0	36	273.0	23	259.1	25
AFG_1_	5.67	329.2	56	243.1	26	283.1	42
AFB_2_	6.08	315.1	60	287.1	26	259.1	30
AFB_1_	6.46	313.2	58	241.1	34	285.1	22
DAS	6.75	384.3	23	307.1	11	229.0	15
HT-2	8.06	442.2	20	215.2	13	263.1	13
OTB	8.42	370.2	31	205.1	16	324.1	13
FB_1_	8.61	722.5	45	352.3	32	334.2	35
T-2	8.90	484.2	20	305.2	15	185.1	25
OTA	9.10	404.2	20	105.1	18	239.1	21
FB_3_	9.48	706.5	68	336.2	35	318.2	35
ZEN	9.50	319.2	34	187.1	20	185.1	25
ST	9.77	325.1	51	310.1	23	281.1	31
FB_2_	10.32	706.4	53	354.2	34	688.2	29
OTC	11.35	432.2	34	239.1	22	358.1	14
[^13^C_17_]-AFB_1_(IS)	6.46	330.2	19	301.1	27	/	/
[^13^C_34_]-FB_1_(IS)	8.61	756.7	50	356.4	40	/	/
[^13^C_20_]-OTA(IS)	9.10	424.4	28	250.0	27	/	/

**Table 2 toxins-14-00742-t002:** Matrix effects of selected mycotoxins in different samples.

Mycotoxins	WBL		BLS		JBL		DFP	
	ME (%)	RSD (%)	ME (%)	RSD (%)	ME (%)	RSD (%)	ME (%)	RSD (%)
DON	78.30	9.88	75.23	7.60	90.04	6.65	95.57	8.09
DOM-1	101.07	7.70	98.33	10.34	83.56	9.87	79.97	7.49
NEO	128.25	12.25	134.78	8.24	100.66	5.37	119.15	3.59
15-AcDON	78.98	5.27	86.50	6.86	90.01	6.13	115.82	10.15
3-AcDON	90.20	5.03	94.28	2.96	84.63	4.93	106.96	2.72
AFM_2_	100.93	6.77	102.75	5.09	113.10	4.02	114.20	11.71
AFG_2_	77.91	3.85	73.85	9.10	79.41	3.03	63.83	7.62
AFM_1_	104.53	8.57	107.21	3.99	108.63	6.56	104.95	7.23
AFG_1_	71.68	8.70	75.97	4.52	75.36	7.91	72.28	12.18
AFB_2_	91.18	9.97	93.93	6.40	102.46	7.29	100.19	2.07
AFB_1_	109.10	9.54	105.93	5.32	121.08	7.03	121.46	11.21
DAS	124.47	7.83	126.74	4.93	120.10	6.23	117.74	6.15
HT-2	87.07	8.20	79.13	4.41	96.54	5.07	106.59	7.52
OTB	101.67	8.48	112.47	8.63	100.44	11.10	109.55	10.29
FB_1_	77.11	8.00	81.49	6.31	79.01	3.75	56.50	3.80
T-2	94.45	6.08	96.82	6.87	91.95	6.01	103.20	3.83
OTA	104.73	5.52	109.08	3.94	110.97	6.10	101.18	2.44
FB_3_	81.61	4.30	87.63	5.89	81.44	5.32	59.14	5.15
ZEN	90.79	4.68	103.88	10.51	106.08	5.99	111.18	8.11
ST	94.35	10.71	92.77	5.41	97.55	5.47	105.63	9.50
FB_2_	84.93	6.48	90.66	3.76	80.96	6.57	66.74	4.63
OTC	108.44	6.67	107.11	8.85	109.17	9.00	111.53	6.21

**Table 3 toxins-14-00742-t003:** Regression equation, correlation coefficients, linearity ranges, LOD and LOQ, and precision for mycotoxins.

Mycotoxins	Calibration Curves	*r* ^2^	Linear Range (μg/kg)	LOQ (μg/kg)	LOD (μg/kg)	Precision (%)	
						Intra-Day (*n* = 6)	Inter-Day (*n* = 6)
DON	*Y* = 0.0097*X* − 0.0118	0.9989	2.5–250	2.5	1.0	4.76	5.75
DOM-1	*Y* = 0.0120*X* − 0.0202	0.9962	2.5–250	2.5	1.0	2.30	4.01
NEO	*Y* = 0.0534*X* − 0.0203	0.9992	2.5–500	2.5	0.8	2.67	2.48
3-AcDON	*Y* = 0.0588*X* − 0.0047	0.9983	2.5–500	2.5	0.8	2.34	2.79
15-AcDON	*Y* = 0.0280*X* − 0.0373	0.9988	2.5–500	2.5	0.8	2.44	3.76
AFM_2_	*Y* = 0.3726*X* − 0.0174	0.9990	0.1–50	0.1	0.04	2.21	3.19
AFG_2_	*Y* = 0.3746*X* − 0.0226	0.9994	0.1–25	0.1	0.05	4.77	5.14
AFM_1_	*Y* = 0.5129*X* − 0.0005	0.9998	0.1–50	0.1	0.04	2.42	3.73
AFG_1_	*Y* = 0.7509*X* − 0.0888	0.9979	0.1–50	0.1	0.04	1.82	2.45
AFB_2_	*Y* = 0.4713*X* − 0.0034	0.9952	0.1–25	0.1	0.05	4.35	7.51
AFB_1_	*Y* = 0.5695*X* − 0.0376	0.9992	0.1–50	0.1	0.04	1.44	2.83
DAS	*Y* = 0.1918*X* + 0.0294	0.9998	2.5–500	2.5	0.8	1.95	2.95
HT-2	*Y* = 0.0134*X* − 0.0194	0.9978	1.0–250	1.0	0.4	3.78	3.96
OTB	*Y* = 0.7845*X* + 0.0042	0.9997	0.25–250	0.25	0.1	1.19	2.04
FB_1_	*Y* = 0.0321*X* − 0.02	0.9970	5.0–1000	5.0	1.5	3.49	4.05
T-2	*Y* = 0.0704*X* − 0.0296	0.9978	1.0–250	1.0	0.4	2.75	3.57
OTA	*Y* = 0.2748*X* + 0.0149	0.9988	0.25–250	0.25	0.1	2.33	4.32
FB_3_	*Y* = 0.0956*X* − 0.0588	0.9991	5.0–1000	5.0	1.5	2.16	4.52
ZEN	*Y* = 0.0247*X* − 0.01	0.9980	1.0–250	1.0	0.4	2.33	4.47
ST	*Y* = 0.4391*X* + 0.3946	0.9997	1.0–250	1.0	0.3	1.80	3.19
FB_2_	*Y* = 0.0648*X* − 0.0226	0.9997	5.0–1000	5.0	1.5	2.44	3.98
OTC	*Y* = 1.5487*X* + 0.0199	0.9954	0.25–250	0.25	0.1	3.29	4.79

**Table 4 toxins-14-00742-t004:** Recoveries of selected mycotoxins in BLS.

Mycotoxins	Spike Levels					
	Low *		Middle		High	
	Recovery (%)	RSD (%)	Recovery (%)	RSD (%)	Recovery (%)	RSD (%)
DON	59.52	6.32	75.38	5.54	71.58	6.89
DOM-1	62.07	8.35	67.13	7.67	76.68	7.33
NEO	109.77	11.13	110.04	10.57	117.28	4.88
3-AcDON	84.97	4.72	86.88	4.46	88.24	5.37
15-AcDON	86.20	8.36	86.04	5.52	81.73	5.83
AFM_2_	87.77	7.07	95.50	6.08	85.96	10.72
AFG_2_	88.98	9.89	87.59	5.01	86.44	5.76
AFM_1_	85.66	7.29	92.47	3.29	93.57	4.35
AFG_1_	79.82	3.29	77.22	5.59	73.65	4.52
AFB_2_	94.09	8.49	98.84	6.00	92.18	7.44
AFB_1_	103.46	4.42	100.41	2.42	95.00	2.86
DAS	103.24	4.28	113.21	5.51	110.13	3.16
HT-2	87.71	4.59	83.55	8.21	90.68	4.80
OTB	102.54	10.81	94.90	5.84	97.17	6.87
FB_1_	86.22	13.31	95.98	11.56	83.54	4.09
T-2	99.53	4.10	97.49	4.40	99.09	3.14
OTA	83.87	14.67	105.47	10.43	106.85	3.56
FB_3_	106.30	3.93	92.59	6.67	90.26	2.77
ZEN	87.44	4.67	96.68	4.17	84.72	8.04
ST	96.58	3.73	105.24	3.71	93.62	2.52
FB_2_	107.77	7.96	103.41	3.26	94.39	3.30
OTC	105.57	5.57	102.69	3.56	98.21	5.85

* Low level, 1.0 μg/kg for AFB_1_, AFB_2_, AFG_1_, AFG_2_, AFM_1_, AFM_2_; 5 μg/kg for OTA, OTB, OTC; 10 μg/kg for DON, DOM-1, HT-2, T-2, ST, ZEN, 15-AcDON, 3-AcDON NEO and DAS; 25 μg/kg for FB_1_, FB_2_, FB_3_; Middle level, five times of low level; High level, twenty-five times of low level.

**Table 5 toxins-14-00742-t005:** Occurrence and levels of selected mycotoxins in samples.

Sample		Mycotoxin Contents (μg/kg)						
P.N/S.N ^a^	Positive	AFB_1_	AFB_2_	AFG_1_	AFG_2_	AFM_1_	AFM_2_	ST
2/15	WBL-4	5.43	2.78	/ ^c^	/	/	Tr	/
	WBL-11	Tr ^b^	/	/	/	/	/	/
13/30	BLS-1	Tr	/	Tr	/	/	/	Tr
	BLS-2	7.55	2.63	/	/	1.36	/	/
	BLS-6	0.39	/	0.17	/	Tr	/	Tr
	BLS-9	Tr	/	Tr	/	/	/	/
	BLS-11	Tr	/	Tr	/	/	/	/
	BLS-13	1.39	Tr	Tr	/	/	Tr	1.10
	BLS-17	1.38	0.20	Tr	/	/	/	/
	BLS-21	0.26	/	0.14	/	/	/	1.38
	BLS-23	0.93	Tr	Tr	/	/	/	2.17
	BLS-24	0.36	0.16	0.05	/	Tr	/	/
	BLS-26	1.62	0.58	0.32	0.15	1.06	/	/
	BLS-27	Tr	/	/	/	/	/	/
	BLS-29	Tr	/	/	/	/	/	/
6/15	JBL-6	Tr	0.13	/	/	/	/	/
	JBL-8	0.33	0.39	Tr	Tr	/	/	/
	JBL-9	Tr	Tr	/	/	/	/	/
	JBL-11	4.02	0.57	Tr	/	/		
	JBL-13	Tr	/	/	/	/	/	/
	JBL-15	1.15	0.28	0.24	0.33	Tr	/	/

^a^ P. N. Positive number; S. N., Sample number. ^b^ Tr: <LOQ; ^c^ /, not detected.

## Data Availability

Not applicable.

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
