# Peer review of "Centrifugation-Assisted Solid-Phase Extraction Coupled with UPLC-MS/MS for the Determination of Mycotoxins in ARECAE Semen and Its Processed Products"

_toxins, 2022, doi:10.3390/toxins14110742_

Round 1

Reviewer 1 Report

This paper is very well written. All aspects of the validation study were taken into account. Only the items listed below should be corrected.

ABSTRACT

Please insert the whole name before for the first time write abbreviation.

“Sterigmatocystin was detected in five Arecae Semen slices”-If You write the number of contaminated samples, You should also write the number of analyzed samples.

“None of the investigated mycotoxins were detected in Arecae pericarpium (Dafupi).” This sentence is not clear before the abstract did not provide information about investigated mycotoxins, which mycotoxins were analyzed?

“Arecae Semen is easily susceptible to fungal infections that produce toxins, and the  resulting safety problems have attracted attention.” Based on what fact You wrote this sentence since analyzed samples in this paper were not frequently contaminated.

From how many different regions Arecae pericarpium were collected? You wrote different regions in Abstract, but it is not clear how many different regions were included in this study.

INTRODUCTION

Abbrevation- please write the full name when for the first time you mention something in the text and then introduce the abbreviation. Please check it in the whole Manuscript. Also, it is not necessary to write the whole name of something if you already write abbrevation (for example UFLC-MS/MS, ACN)

“stengmatocystin” is this sterigmatocystin?

“67.13, 111.21, 13.33%.” –Please reduce the number of decimal digits.

Why You wrote Tr as < LOD?

Author Response

Q1. Please insert the whole name before for the first time write abbreviation.

Response: Thanks for your suggestion! We have made modifications in the revised manuscript according to the reviewers’ comments.

Q2.“Sterigmatocystin was detected in five Arecae Semen slices”-If You write the number of contaminated samples, You should also write the number of analyzed samples.

Response: Thanks! We have added the number of Arecae Semen slices in the abstract.

Q3. “None of the investigated mycotoxins were detected in Arecae pericarpium (Dafupi).” This sentence is not clear before the abstract did not provide information about investigated mycotoxins, which mycotoxins were analyzed?

Response: Thanks! We have added the investigated mycotoxins in the abstract.

Q4. “Arecae Semen is easily susceptible to fungal infections that produce toxins, and the resulting safety problems have attracted attention.” Based on what fact You wrote this sentence since analyzed samples in this paper were not frequently contaminated.

Response: Thanks! We have made modifications in the revised manuscript

Q5. From how many different regions Arecae pericarpium were collected? You wrote different regions in Abstract, but it is not clear how many different regions were included in this study.

Response: Thanks for your suggestion! The information of sample collection regions has been supplemented in the section 4.3.

Q6. Abbrevation- please write the full name when for the first time you mention something in the text and then introduce the abbreviation. Please check it in the whole Manuscript. Also, it is not necessary to write the whole name of something if you already write abbrevation (for example UFLC-MS/MS, ACN)

Response: Thanks for your suggestion! We have made modifications and checked it in the whole revised manuscript.

Q7. “stengmatocystin” is this sterigmatocystin?

Response: Yes. Thank you for your careful review. We have revised it in the manuscript.

Q8.  “67.13, 111.21, 13.33%.” –Please reduce the number of decimal digits.

Response: Thanks. We have reduced the number of decimal digits in the manuscript.

Q9. Why You wrote Tr as < LOD?

Response: We have made modifications in the revised manuscript.

Reviewer 2 Report

See attached below

Author Response

Response: Thanks for your careful review. We have added the word “not” between does and accumulate in this sentence of “The aflatoxinogenic species does not accumulate sterigmatocystine”

Reviewer 3 Report

- in order to conclude that this method can be applied to the determination of mycotoxins on herbs, a validation is also necessary on these matrices.

-in subsection 4.1. a fragment appears that I don't think is related to the research carried out

Author Response

Q1. - in order to conclude that this method can be applied to the determination of mycotoxins on herbs, a validation is also necessary on these matrices.

Response: Thanks for your suggestions, which will help us improve the quality of the article. Matrix effect and recovery are two important parameters to investigate the applicability of the developed method on different herbs. In the optimization of methodology, we investigated the recovery and matrix effect on WBL, BLP, JBL and DFP. The results of recovery rate have been added in the supplementary materials (Table S1).

Q 2. -in subsection 4.1. a fragment appears that I don't think is related to the research carried out

Response: Thanks for your careful review. This fragment is a paragraph in the Toxins submission template. We have deleted it in the revised manuscript.